# Vaccination with *Neospora* GRA6 Interrupts the Vertical Transmission and Partially Protects Dams and Offspring against *Neospora caninum* Infection in Mice

**DOI:** 10.3390/vaccines9020155

**Published:** 2021-02-15

**Authors:** Ragab M. Fereig, Hanan H. Abdelbaky, Yoshifumi Nishikawa

**Affiliations:** 1National Research Center for Protozoan Diseases, Obihiro University of Agriculture and Veterinary Medicine, Inada-cho, Obihiro, Hokkaido 080-8555, Japan; ragabfereig2018@gmail.com (R.M.F.); hananragabegypt@gmail.com (H.H.A.); 2Department of Animal Medicine, Faculty of Veterinary Medicine, South Valley University, Qena City, Qena 83523, Egypt

**Keywords:** *N. caninum*, vaccine, neosporosis, NcGRA6, abortion

## Abstract

Vaccination is the mainstay of preventative measures for numerous infectious diseases. *Neospora caninum* infection induces storms of abortion in pregnant cows and ewes, resulting in drastic economic losses because of fetal losses and culling of the dams. Herein, we evaluated the potential of recombinant protein of *N. caninum* dense granule protein 6 fused with glutathione-*S*-transferase (NcGRA6+GST) as a vaccine candidate against neosporosis in a pregnant mouse model. The protective efficacy was investigated by subcutaneous inoculation of BALB/c mice with recombinant NcGRA6+GST (25 pmol), and GST alone (25 pmol) or phosphate-buffered saline (PBS) as the controls. This study revealed the partial ability of NcGRA6+GST to protect the dams and offspring from *N. caninum* infection during the critical period of pregnancy. This ability was revealed by higher survival rate and lower parasite burden in brains of offspring of the NcGRA6+GST-immunized group in comparison with the control groups. In addition, mouse dams from NcGRA6+GST-immunized groups exhibited lower clinical score and minimum alteration in body weight in comparison with PBS or GST groups after challenge with *N. caninum* tachyzoites. Taken together, our results suggest the efficacy of recombinant NcGRA6 for interrupting the vertical transmission of *N. caninum* in mice by reducing the severity of infections in dams and offspring.

## 1. Introduction

*Neospora caninum* is a protozoan parasite that infects a wide range of animal species. It induces storms of abortion in pregnant cows and ewes, resulting in drastic economic losses. Dogs, as definitive hosts, are severely infected by *N. caninum* and have exhibited paralysis of hind limbs. Neosporosis has been reported in many countries and in different regions of the world [1]. Although there is no evidence for neosporosis in humans, the disease recently gained significant interest because of the substantial economic losses associated with the abortion of cattle [2]. *N. caninum* is one of the most successful pathogens in crossing the blood–placenta barrier at the feto–maternal interface. As a result, infected pregnant cows may abort at any stage of gestation, with most abortions occurring at five to six months. Fetuses may die in uteri, be born clinically infected, or be clinically normal but congenitally infected. Congenitally infected calves remain persistently infected and can pass the infection onto their offspring [1,2].

In a systematic review, the median estimate of the worldwide economic losses of *N. caninum*-infections or abortions was calculated to be USD 1.3 billion per annum [3]. Currently, no commercially effective vaccines or licensed pharmacological treatments are available for bovine neosporosis. Under problematic treatment caused by the complex life cycle of the parasite, strategic vaccination protocol is the most promising tool for *Neospora* control [4].

For *Neospora* immunological control, humoral and cellular immunities including a balance of T-helper 1 (Th1) and T-helper 2 (Th2) activity are a critical aspect. In this regard, IFN-γ was recognized as the most powerful molecule combating *N. caninum* infection via several pathways. Moreover, the specific antibodies and IL-4 as mediators of Th2-type immune response exert a regulatory effect for protection against neosporosis [5,6]. Many successful vaccine trials against *N. caninum* infection in pregnant mouse model have been reported. In general, the live attenuated vaccine has the most significant effect and the predominantly successful vaccine for protection of mice against transplacental neosporosis [7,8,9]. Killed vaccine has also been evaluated and induced a marked protective effect [10]. In the same context, many subunit vaccines have variably protected mice against transplacental transmission of *N. caninum*, including microneme proteins 1 and 2 combined with rhoptry protein 1 [11], dense granule 7 [12], dense granule 6 [13], apical membrane antigen 1 [14], and rhoptry proteins 2 and 40 [15]. However, although it was removed from the market, the only approved vaccine for commercial use for prevention of vertical transmission of neosporosis in cattle (Neoguard) was prepared from tachyzoite lysate antigen [16].

Peculiarly, cumulative evidence has suggested that dense granule antigens possessing the capability to interact efficiently with the host immune system with different approaches. In mice, immunization with NcGRA7 encapsulated in oligomannose-coated liposomes (NcGRA7-OML) was reported to reduce the congenital infection of *N. caninum* via eliciting Th1-mediated immunity and antibody production [12]. In a similar pathway as reported in mice, NcGRA7-OML reduced the severity of *N. caninum* infection in immunized calves [17]. In a more recent study, NcGRA7 was found to play a remarkable role in pathogenesis of neosporosis in mice via modulating host immunity using knockout parasite [18]. *N. caninum* cyclophilin was also reported to protect significantly BALB/c and C57BL/6 mice, particularly when encapsulated in OML. This protection was highly dependent on toll-like receptor 2 when C57BL/6 mice and TLR2-knockout mice were investigated [19].

The potential of NcGRA6 as a vaccine candidate has also been investigated against *N. caninum* infection in non-pregnant and pregnant mouse models. As a vector-based vaccine, NcGRA6 expressed in *Brucella abortus* (strain RB51) triggered protective immunity against lethal intraperitoneal infection and vertical transmission of *N. caninum* infection [13,20]. However, safety concerns about the hazards of *B. abortus* RB51 is a major limiting aspect for further application of such an approach. Accordingly, in our previous study [21], we investigated the efficacy of recombinant protein of NcGRA6 with and without OML adjuvant as a vaccine antigen in a non-pregnant mouse model. We revealed the immune-stimulatory effect of NcGRA6 as evidenced by induction of IL-12p40 secretion from murine macrophage. Moreover, significant protective efficacy of NcGRA6 was reported against non-pregnant BALB/c mice challenged with lethal dose of Nc-1 tachyzoites. Protective effect and triggered immunity of naked NcGRA6 was superior to NcGRA6 formulated with OML adjuvant. NcGRA6 could elicit humoral and cellular immune response. Thus, in the current study, we investigated the potential of naked NcGRA6 as a vaccine candidate against neosporosis in a pregnant mouse model. Indeed, vaccination of pregnant mice with NcGRA6 alone demonstrated promising results for protection against vertical transmission of neosporosis.

## 2. Materials and Methods

### 2.1. Ethics Statement

Procedures that may result in pain or distress of mice in this study were performed under general anesthesia with isoflurane. We followed the guidelines and recommendations of the Guide for the Care and Use of Laboratory Animals of the Ministry of Education, Culture, Sports, Science and Technology, Japan. The procedures were approved by the Committee on the Ethics of Animal Experiments at the Obihiro University of Agriculture and Veterinary Medicine (permission numbers 29–58, 29–64, 18–44, 18–50, 19–56).

### 2.2. Animals

Female and male BALB/c mice aged 6–7 weeks were purchased from Clea Japan (Tokyo, Japan). The mice were housed under specific pathogen-free conditions in the animal facility of the National Research Center for Protozoan Diseases at Obihiro University of Agriculture and Veterinary Medicine, Obihiro, Japan.

### 2.3. Parasites and Cell Cultures

*N. caninum* (Nc-1) isolate was maintained in Vero cells (African green monkey kidney epithelial cells) cultured in Eagle’s minimum essential medium (EMEM; Sigma, St. Louis, MO, USA) containing 8% heat-inactivated fetal bovine serum (FBS; Nichirei Biosciences, Tokyo, Japan) and 1% streptomycin–penicillin (Sigma). The parasites were purified; host cell debris was washed in cold phosphate-buffered saline (PBS); and the infected cell monolayer was scraped with a cell scraper (BD Bioscience, San Jose, CA, USA), harvested in medium, and centrifuged at 800× *g* for 5 min at 20 °C. The final cell pellet was resuspended in Roswell Park Memorial Institute (RPMI) 1640 medium (Sigma) and passed through a 27-gauge needle and a filter with a pore size of 5.0 µm (Millipore, Bedford, MA, USA).

### 2.4. NcGRA6 Gene Amplification and Cloning, and Expression of Recombinant Proteins

Cloning of *NcGRA6* gene and expression of recombinant proteins of *N. caninum* dense granule protein 6 fused with glutathione-*S*-transferase (NcGRA6 + GST) and glutathione-*S*-transferase (GST) were performed as previously described [21]. In brief, the PCR products digested with *Eco*RI and *Xho*I were inserted into the pGEX-4T1 plasmid vector (Amersham Pharmacia Biotech, Madison, CA, USA). Recombinant NcGRA6 was expressed as glutathione *S*-transferase (GST) fusion protein (NcGRA6+GST) in *Escherichia coli* BL21 (DE3) (New England BioLabs Inc., Ipswich, MA, USA). The purity and amount of the proteins were determined by running sample on sodium dodecyl sulfate polyacrylamide gel electrophoresis (SDS-PAGE) followed by staining with Coomassie Brilliant Blue R250 (MP Biomedicals Inc., Illkirch-Graffenstaden, France). The proteins were dialyzed in PBS and filtered with a 0.45-μm low-protein binding Supor membrane, and the endotoxin was removed using Acrodisc Units with Mustang E Membrane (Pall Life Sciences, Ann Arbor, MI, USA) for subsequent use for in vivo experiments. In addition, level of endotoxin was measured with limulus amebocyte lysate reagents (Seikagaku Inc., Tokyo, Japan), and no endotoxin was found in the tested protein lots.

### 2.5. Scheme for Mice Immunization, Mating, and Infection

All female mice were immunized via subcutaneous inoculation with recombinant proteins of NcGRA6+GST (25 pmol), GST (25 pmol), or PBS alone (100 µL) 3 times in 14-day intervals. Seven days after the third immunization, the female mice (11 weeks old) were housed with male mice for 4 days (2 females with 1 male per cage) and checked twice daily for the presence of seminal plugs in the vagina. The first day on which a plug was noticed was designated as day 0 of gestation for each individual. At 4 weeks post-vaccination, only pregnant dams were challenged intraperitoneally on the same day with 1 × 10^5^ tachyzoites of *N. caninum* (Nc-1) at day 10 of gestation (trial 1: PBS: *n* = 3, GST: *n* = 3, NcGRA6+GST: *n* = 4; trial 2: PBS: *n* = 5, GST: *n* = 6, NcGRA6+GST: *n* = 6). Offspring numbers and survival rates were measured daily until 30 days after birth. The brain and uterus of all dams and the brain of some mice that succumbed to infection and all surviving offspring at 30 days after birth (correspondent to 40–42 days post-infection) were aseptically collected to determine the parasite burden. Mice were considered as pregnant if they showed vaginal plug and marked increase in body weight during the first term of pregnancy (>20% from day 0 of weighing).

### 2.6. Real-Time PCR for Measurement of Parasite Burden

The parasites DNA from the brain and uterus of dam and from brain of offspring mice were extracted, purified, and quantified. Homogenate from tissues was obtained after lysis with extraction buffer (0.1 M Tris-HCl (pH 9.9), 1% SDS, 0.1 M NaCl, 1 mM Ethylenediaminetetraacetic acid (EDTA), 1 mg/mL proteinase K) at 50 °C. Phenol–chloroform extraction and ethanol precipitation were used to purify the DNA. Parasite DNA was counted and analyzed by real-time PCR using specific primers for Nc5 gene, the forward primer: 248–257 nucleotides (5′-ACT GGA GGC ACG CTG AAC AC-3′) and the reverse primer: 303–323 nucleotides (5′-AAC AAT GCT TCG CAA GAG GAA-3′). The PCR was performed on genomic DNA and guided by the SYBR green detection method (Applied Biosystems, Carlsbad, CA, USA). Amplifications, data processing and analysis were performed with the ABI Prism 7900HT Sequence Detection System (Applied Biosystems). A standard curve was established with *N. caninum* DNA extracted from 1 × 10^5^ parasites using 1 μL of a serial dilution ranging from 10,000 to 0.01 parasites. The data were analyzed using Dissociation Curves version 1.0 F (Applied Biosystems).

### 2.7. Clinical Score and Body Weight

Monitoring and analyses of changes in body weight and clinical score of each individual mouse were recorded daily for infected mice and compared with the same one at the first day of measurement, as described in our previous study [21]. Daily observations were applied from 2 days before mating of mice to 30 days after birth (40–42 dpi), because of variability in birth days of different dams (18–20 after conception), except for the peripartum period (−2 to 5 days after delivery) to exclude the pregnancy parturition effects and avoid the disturbances of mice, where only recording of the offspring numbers was performed.

### 2.8. Statistical Analysis

All statistical analyses were performed with GraphPad Prism version 5 (GraphPad Software Inc., La Jolla, CA, USA). One-way analysis of variance (ANOVA) followed by the Tukey–Kramer test were used for group comparisons of parasite burden. Two-way ANOVA followed by the Bonferroni posttest was used for estimating differences in clinical score and body weight results; data are presented as a standard error of the mean. The differences among birth rates of different groups were tested using a χ^2^ test. The significance of the differences in survival rates of offspring was analyzed with a log-rank test. The levels of statistical significance are presented with asterisks (*) or number sign (#) and are defined in each figure legend. A *p*-value < 0.05 was considered statistically significant unless stated elsewhere.

## 3. Results

### 3.1. Protective Efficacy of NcGRA6 Immunization in Mouse Offspring

Table 1 shows the outcome of *N. caninum* infection in dams and offspring in both trials. Analyzing both trials, no significant difference in birth rate of pregnant dam was recorded in NcGRA6+GST-immunized mice (10/10; 100%) in comparison with those of PBS (7/8; 87.5%) and GST (8/9; 88.9%) groups (*p* = 0.249, NcGRA6+GST vs. PBS; *p* = 0.278, NcGRA6+GST vs. GST). The numbers of offspring per litter in the group receiving immunization with NcGRA6+GST (M ± SD; 5.2 ± 1.2) were not different from those in the group receiving PBS (5.7 ± 2.2) or GST (7.3 ± 1.3). Next, the survival rates of offspring were examined for 30 days after birth. The survival rate for the offspring from dams immunized with NcGRA6+GST (15/52; 28.8%) was significantly higher than those of PBS and GST groups (*p* < 0.0001), where all offspring were succumbed (PBS, 0/40; GST, 0/57) (Table 1 and Figure 1).

When analyzing trial 1 and trial 2 separately for survival rates of offspring, we found that the tendency showed similar effect concerning higher survivals of the NcGRA6 + GST-immunized group than control groups (trial 1: PBS: 0/14, GST: 0/12, NcGRA6+GST: 8/22 (36.4%); trial 2: PBS: 0/26, GST: 0/45, NcGRA6 + GST: 7/30 (23.3%)), with high levels of significance (trial 1: *p* = 0.0003, NcGRA6+GST vs. PBS; *p* = 0.0002, NcGRA6+GST vs. GST; trial 2: *p* = 0.0006, NcGRA6+GST vs. PBS; *p* = 0.0005, NcGRA6 + GST vs. GST). Regarding parasite burden, brain of offspring of immunized dams with NcGRA6 + GST showed the lowest parasite number in relation to other groups (PBS and GST), particularly in ones that survived, although the difference was not significant (*p* > 0.05) (Figure 2). This result suggests that immunization of mice with NcGRA6 induced partial reduction of vertical transmission of neosporosis in pregnant BALB/c mice.

### 3.2. Protection Indices in NcGRA6-Immunized Mouse Dams

The survival rate of dams did not show significant differences among different groups (*p* > 0.05) (Figure 1). While it was not significant, the parasite burden was lower in the brain of dams of the NcGRA6+GST-immunized group than those of PBS or GST groups (*p* > 0.05) (Figure 2). On the contrary to PBS- and GST-inoculated mouse groups, the parasite number in uterine tissue of NcGRA6+GST-immunized dams was undetectable trial 1. In trial 2, parasite burden of NcGRA6+GST-immunized mice was similar to GST group and was lower than the PBS group, although the difference was not statistically different (*p* > 0.05) (Figure 2). For further investigation, the clinical score and body weight were monitored and measured daily for all mice in each group. The NcGRA6+GST-immunized dams exhibited lower clinical score after the delivery compared to PBS and GST groups (Figure 3).

Moreover, lower alteration in body weight was reported in NcGRA6+GST-immunized groups in comparison with the controls (PBS or GST), although it was not a statistically significant difference (*p* > 0.05) (Figure 4). The average of each group also indicated the development of pregnancy and the stability of body weight in dams from NcGRA6+GST immunized mice in comparison with other groups (Figure 4). No differences were recorded among the PBS- and GST-inoculated mouse groups, either in clinical score or in the body weight (*p* > 0.05) (Figure 3 and Figure 4). These results indicated that immunization with the naked recombinant protein of NcGRA6 protected dams against *N. caninum* infection during pregnancy.

## 4. Discussion

*Neospora caninum* is incriminated as a major cause of abortion in cattle worldwide, and thus represents a cause of high economic losses of cattle industry. Vaccination is regarded as the ideal control strategy in cases of high prevalence of the disease [22]. In the last decade, tremendous advances in vaccine development against *N. caninum* were achieved, including the discovery of potent vaccine candidates and technologies. However, the limited success on farm animals and few vaccine candidates that protect against vertical transmission rendering the current *Neospora* vaccine development remains largely unsuccessful.

Although the immune system of mice and cattle differs greatly, the mice are regarded as a valuable model for *Neospora* infection studies, and ought to be tested in preliminary screening of efficient vaccine candidates. Mouse models are highly beneficial in recognition of the key effector molecules of innate and adaptive immune responses against *N. caninum* infection. Indeed, using of different mouse models has greatly enhanced our understanding of *N. caninum*–host interactions and assisted in discoveries of potent vaccine candidates [5,6,16]. In cattle, there are many pathways that are responsible for the control of infection and for the preventing of the conversion from tachyzoites to bradyzoites. Natural killer cell, and CD4^+^ and CD8^+^ T cells are involved in the *Neospora* control by destruction of infected cells. Combating of parasite and reducing the multiplication can be achieved via the inflammatory cytokines IFN-γ and TNF-α. Furthermore, humoral immune response exerts a control approach via blocking of the parasite entry into host cells through antibodies [23,24]. In the case of mice, IL-12 is a key cytokine in control of *N. caninum* through enhancement of IFN-γ production. In addition, CD4^+^ but not CD8^+^ T cells are essential in combating *N. caninum* infection in mice. Although these results indicate superiority of a Th1-type immune response for protection, it is more likely that an appropriate Th1/Th2 balance is required [16,25,26].

In our previous study employing a non-pregnant mouse model, we indicated that recombinant NcGRA6 possessed an immune-stimulatory effect. In addition, naked NcGRA6+GST showed higher protective immunity and protection than NcGRA6+GST combined with OML adjuvant [21]. Accordingly, in the present study, we investigated the protective potential of naked NcGRA6 as a vaccine candidate using a pregnant BALB/c mouse model. Herein, we demonstrated the high capacity of NcGRA6 in the reduction of vertical transmission of *N. caninum* in mice evidenced in higher survival rates and lower parasite burden in surviving offspring and dams compared with control groups. Some newborn mice died from the NcGRA6+GST-immunized group and PCR measurements revealed their positivity to *N. caninum* infection. However, the significant higher survival rate and lower parasite burden in the brain of surviving offspring indicated the adequacy of recombinant NcGRA6 immunization in minimizing the vertical transmission.

In spite of using different mouse strains and *N. caninum* isolates, most of the studies revealed that no vaccine candidates could induce sterile immunity or completely prevent the vertical transmission from dams to offspring, especially if subunit vaccines were used. Immunization of Qs mice with recombinant antigens MIC10 and p24B caused a reduction in parasite transmission from dams to offspring by 13.2% and 7.8%, respectively. In the group of mixed injection with MIC10 and 24B, higher reduction of transmission was observed (32.9%) [7]. In a BALB/c pregnant mouse model, the highest survival rates were observed for the groups receiving recombinant NcROP2 (50%) and recombinant NcROP2/NcMIC1/NcMIC3 (35%), while all pups died in the infection and adjuvant control groups [11]. In another study, immunization of BALB/c mice with NcGRA7-OML induced higher survival rates in pups (68.6%) compared to PBS (17.5%) or GST-OML (18.6%) control groups [12]. Immunization of female C57BL/6 mice with recombinant strain RB51 of *Brucella abortus*-expressing NcGRA6 increased the pup survival to 90.7% compared to non-vaccinated mice (81.5%) or those who received RB51 only (61.5%) [13]. Conversely, some previous studies that used live attenuated or killed parasites or whole lysate antigens as vaccine reported higher level of protection than subunit vaccines [7,8,9,10]. However, because of safety, ease of preparation, and affordability, subunit vaccines are more advantageous than other kinds of vaccine. Although NcGRA6-conferred protection is partial in this study, this protective effect can be improved by several strategies such as using appropriate adjuvants or combination with other potent antigens.

Previous studies have revealed the different aspects of the host–parasite relationship in pregnant cattle following *N. caninum* infection [27]. The experimental infection of heifer with *N. caninum* in early gestation period induces death of the fetus. Because the mother can induce a strong cell proliferation response with production of IFN-γ against *N. caninum* in early gestation, inflammation at the maternal–fetal interface may trigger the abortion. In addition, infection at mid-gestation modulates the immune response towards the Th2 cytokine environment at the maternal–fetal interface, resulting in the reactivation of the parasites. Thus, parasite invasion of the placenta and infection of the fetus occurs by the vertical transmission [28].

In the case of pregnant mouse model of *N. caninum* infection, experimental infection with *N. caninum* at mid-gestation induces the vertical transmission [12,14,29]. Some previous studies reveal the importance of Th-2-mediated immunity, especially IL-4, in maintaining the pregnancy in mice infected with *N. caninum* and the counterpart effect of IFN-γ [30,31]. Conversely, several other studies reported the favorable effect of IFN-γ in normal pregnancy development. In cases of *Neospora* challenge infection, many vaccine antigens induce IFN-γ production in re-stimulation assay of spleen cells, while these antigens conferred high protection in the vertical transmission model [12,20,32]. In a similar effect but in a non-infected mice model, some reports revealed the necessity of IFN-γ for normal gestational development in mice [33,34]. These findings reflect the intricate immunomodulation during pregnancy and the distinct effect of IFN-γ depending on the stage of pregnancy and site of operation. Cell-mediated immunity at the site of infection is critical to maintain the pregnancy and control the parasite in dams. Killing the intracellular *N. caninum* at the early stage of the infection may regulate the inflammatory reaction at a minimum level. Thus, the significant prophylactic potential of recombinant NcGRA6 against infection with *N. caninum* in both adult and newborn mice may be dependent on the early induction of cellular immune responses. Our previous relevant study in non-pregnant mice showed robust stimulation of NcGRA6 for IFN-γ and splenocyte proliferation in NcGRA6-immunized groups. In addition, specific anti-NcGRA6 IgG1 and IgG2a had been observed only in NcGRA6-immunized groups but not in PBS- or GST-treated groups. Both studies utilized the same immunization regimen (three immunization times within 2 weeks interval), mouse sex and strain (female BALB/c), dose, and route of administered antigen (25 pmol, subcutaneous) for recalling stimulation assay [21]. In addition, similar levels and tendencies for specific anti-NcGRA6 IgG1 and IgG2a were observed in our present study (see Appendix A) and a previous study [21]. Thus, we expect a potent role of IFN-γ and specific antibody to NcGRA6 in protection induced by NcGRA6 immunization in our current pregnant model.

## 5. Conclusions

The vaccine development focusing on interrupting the vertical transmission of *N. caninum* is the most effective and promising strategy for *Neospora* control and therefore the reduction of the economic losses in the livestock industry. Vaccination with recombinant NcGRA6 interrupted the vertical transmission of neosporosis from infected dams to the offspring in a mouse model. In addition, severity of infection in NcGRA6-immunized dams was significantly lower than the control groups. Our current data would be additional evidence for the potential of NcGRA6 as a vaccine candidate in a pregnant animal model. However, additional studies are needed to investigate the mechanism of protective immunity induced by NcGRA6 in pregnant mice and to increase the effects of blocking the vertical transmission by using appropriate adjuvants or combination with other potent antigens for application to farm animals such as cattle and sheep.

## Figures and Tables

**Figure 1 vaccines-09-00155-f001:**
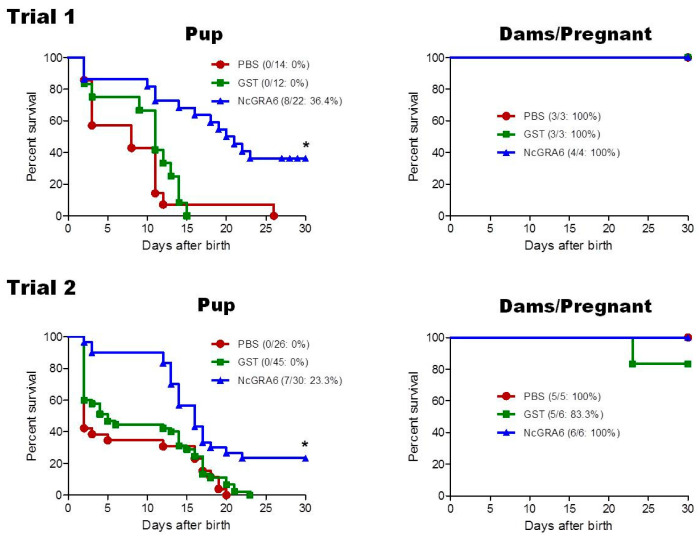
Protective efficacy in offspring and dams from immunized and control groups. Pregnant mice from three groups: PBS (*n* = 3), GST (*n* = 3), and NcGRA6 + GST (*n* = 4) for trial 1, and PBS (*n* = 5), GST (*n* = 6), and NcGRA6 + GST (*n* = 6) for trial 2 were intraperitoneally challenged with non-lethal dose (1 × 10^5^) of *N. caninum* tachyzoites. Upper panel is survival rates of offspring and dams from trial 1, while lower panel is survival rates of offspring and dams from trial 2. The survival rates were calculated until 30 days after birth. Survival curves were generated with the Kaplan–Meier method. Significant differences between NcGRA6+GST-immunized offspring and both control groups according to the log-rank test (*; *p* < 0.001).

**Figure 2 vaccines-09-00155-f002:**
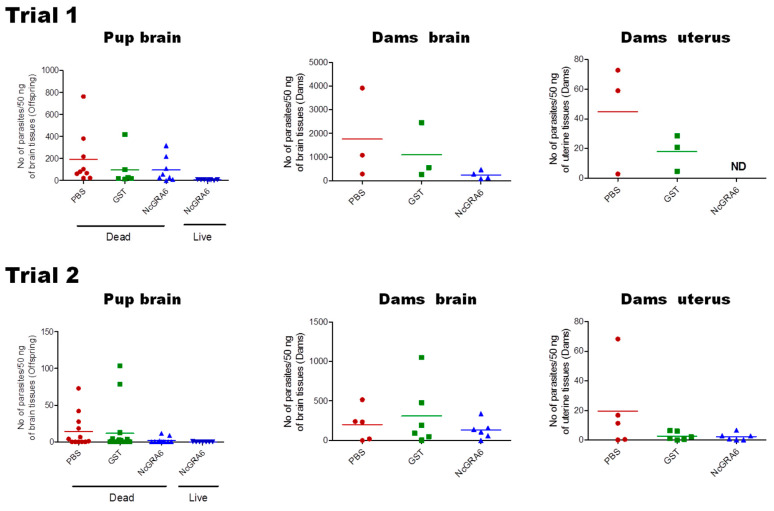
Parasite burden in brain of offspring and brain and uterus of dam. Parasite numbers were measured in the brains of the dead and surviving offspring at 30 days after birth. Some brains of dead offspring could not be collected because of cannibalistic behavior by dam. No statistically significant differences were observed among different groups with one-way ANOVA plus Tukey–Kramer post hoc analysis, *p* > 0.05. In case of dam, pregnant mice were monitored daily until 30 days after birth (40–42 dpi). The results of parasite burden in the brain were analyzed with one-way ANOVA plus a Tukey–Kramer post hoc analysis, and no significant difference was observed between different groups of dams. ND, no detectable number was observed. Horizontal solid lines with different colors represent the mean of calculated values.

**Figure 3 vaccines-09-00155-f003:**
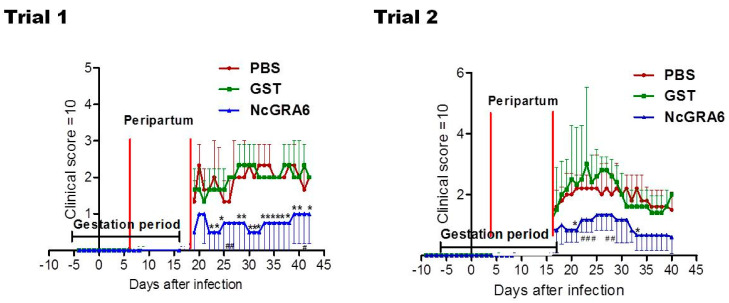
The clinical score observed in pregnant mice. Clinical scores were monitored daily until 30 days after delivery for all mice in each group. The average of all mice in each group was calculated as means ± SD of clinical score values of all mice in a group. The significance of (*) or (#) in the average clinical score was determined with two-way ANOVA plus Bonferroni post hoc analysis (*p* < 0.05) of NcGRA6+GST immunized groups against both PBS and GST groups or against PBS group only, respectively.

**Figure 4 vaccines-09-00155-f004:**
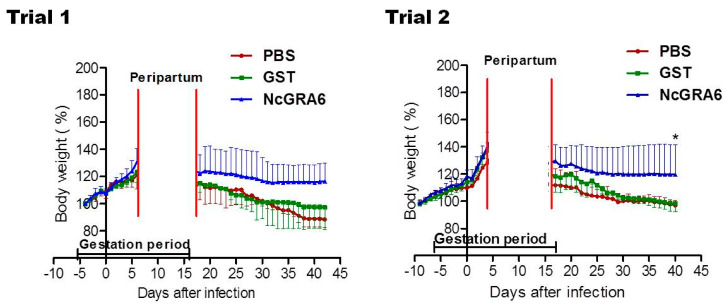
The body weight of pregnant mice. The body weight was monitored daily until 30 days after delivery for all mice in each group. The average of all mice in each group was calculated as means ± SD of body weight values of all adult mice in a group. No significant difference was observed in the body weight (two-way ANOVA and Bonferroni posttest, *p* > 0.05). Pregnancy in all mice was confirmed by visible vaginal plug associated with remarkable increase in body weight (increase in body weight by >20% than that at the day of confirmation of vaginal plug).

**Table 1 vaccines-09-00155-t001:** Protection of *Neospora*
*caninum* dense granule protein 6 (NcGRA6) vaccine against *N. caninum* infection in pregnant BALB/c mouse model.

Group	Experiment	No. of Litters/No. of Pregnant Mice (Birth Rate %) ^#^	Mean No. of Offspring Per Litter (SD)	No. of Surviving Offspring/No. of Offspring in Litter	Total No. of Surviving Offspring/ Total No. of Offspring (%)
PBS	Trial 1	2/3 (66.7)	7 (1.4)	0/8, 0/6	0/14 (0)
	Trial 2	5/5 (100)	5.2 (2.4)	0/6, 0/3, 0/4, 0/4, 0/9	0/26 (0)
	Total	7/8 (87.5)	5.7 (2.2)	-	0/40 (0)
GST	Trial 1	2/3 (66.7)	6 (0)	0/6, 0/6	0/12 (0)
	Trial 2	6/6 (100)	7.5 (1.4)	0/8, 0/9, 0/6, 0/6, 0/7, 0/9	0/45 (0)
	Total	8/9 (88.9)	7.3 (1.3)	-	0/57 (0)
NcGRA6+GST	Trial 1	4/4 (100)	5.5 (1.3)	0/4, 3/6, 2/5, 3/7	8/22 (36.4) *
	Trial 2	6/6 (100)	5 (1.3)	0/4, 0/6, 5/6, 0/6, 2/5, 0/3	7/30 (23.3) *
	Total	10/10 (100)	5.2 (1.2)	-	15/52 (28.8) *

^#^ The mice were considered pregnant when they exhibited visible vaginal plug combined with remarkable increase in body weight (increase in body weight by >20% than that at the day of confirmation of vaginal plug). * The statistically significant difference of pup survival of the NcGRA6+glutathione-*S*-transferase (GST)-vaccinated group against both phosphate-buffered saline (PBS)- and GST-inoculated groups for each trial or total number using a log-rank test.

## Data Availability

The data of this research can be provided based on request from the correspondence author (project supervisor).

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
