# Peer review of "Vaccination with Neospora GRA6 Interrupts the Vertical Transmission and Partially Protects Dams and Offspring against Neospora caninum Infection in Mice"

_vaccines, 2021, doi:10.3390/vaccines9020155_

Round 1
Reviewer 1 Report
The study deals with an interesting topic about how NcGRA6 reduces neosporosis in pregnant mice. The design of the study is clear and the results are well presented.
- Generally, I wish to see in the revised version more evidence about the correlation of NcGRA6 application in the human and beneficial effects. Also, a comparison with the traditional vaccines is required in this case. This can be as one paragraph with relevant data in the introduction and discussion sections.
- Also, I suggest the authors draw the limitation of the present study in the end of the discussion section.
- Please format all the genes abbreviations in the whole text in the Italic form
Author Response
Reviewer 1
- The study deals with an interesting topic about how NcGRA6 reduces neosporosis in pregnant mice. The design of the study is clear and the results are well presented.
Author’s response: We are greatly appreciating the positive comment from the reviewer 1.
- Generally, I wish to see in the revised version more evidence about the correlation of NcGRA6 application in the human and beneficial effects. Also, a comparison with the traditional vaccines is required in this case. This can be as one paragraph with relevant data in the introduction and discussion sections.
Author’s response:
We have added this information regarding N. caninum infection in humans in introduction section;
“Although there is no evidence for neosporosis in humans, the disease recently gained significant interest because of the substantial economic losses associated with the abortion of cattle [2].” (lines 46-48)
In discussion section:
We have compared our NcGRA6 vaccine against other kinds of commonly used vaccines against N. caninum;
“In spite of using different mouse strains and N. caninum isolates, most of studies revealed that no vaccine candidates could induce sterile immunity or completely prevent the vertical transmission from dams to offspring especially if subunit vaccines were used. Immunization of Qs mice with recombinant antigens MIC10 and p24B caused a reduction in parasite transmission from dams to offspring by 13.2% and 7.8%, respectively. In the group of mixed injection with MIC10 and 24B, higher reduction of transmission was observed (32.9%) [7]. In BALB/c pregnant mouse model, the highest survival rates were observed for the groups receiving recombinant NcROP2 (50%) and recombinant NcROP2/NcMIC1/NcMIC3 (35%) while all pups died in the infection- and adjuvant- control groups [11]. In another study, immunization of BALB/c mice with NcGRA7-OML induced higher survival rates in pups (68.6%) compared to PBS (17.5%) or GST-OML (18.6%) control groups [12]. Immunization of female C57BL/6 mice with recombinant strain RB51 of Brucella abortusexpressing NcGRA6 increased the pup survival to 90.7% compared to non-vaccinated (81.5%) or RB51 only (61.5%) receiving groups [13]. Conversely, some previous studies that used live attenuated or killed parasites or whole lysate antigens as vaccine reported higher level of protection than subunit vaccines [7-10].” (lines 311-326)
Also, we referred to NcGRA6 application as a subunit vaccine;
“However, because of safety, ease of preparation and affordability, subunit vaccines are advantageous than other kinds of vaccine. Although NcGRA6-conferred protection is partial in this study, this protective effect can be improved by several strategies such as using appropriate adjuvants or combination with other potent antigens” (lines 326-329)
- Also, I suggest the authors draw the limitation of the present study in the end of the discussion section.
Author’s response:
We have referred the limitation as follows;
“Our current data would be additional evidence for the potential of NcGRA6 as a vaccine candidate in pregnant animal model. However, additional studies are needed to investigate the mechanism of protective immunity induced by NcGRA6 in pregnant mice and to increase the effects of blocking the vertical transmission by using appropriate adjuvants or combination with other potent antigens for application to farm animals such as cattle and sheep.” (lines 370-375)
- Please format all the genes abbreviations in the whole text in the Italic form
Author’s response:
We have confirmed the formatting of gene names to italic, while those referred to protein names were not italicized.

Reviewer 2 Report
The authors present a vaccine candidate to prevent vertical transmission of N. caninum. N. caninum infection of cattle results in nearly $1.3b annual loss to the cattle industry and is thus an economically important pathogen. Vaccines are necessary to prevent disease. The authors show in a murine model, reduction in pathogen burden weight loss, and pup survival, however the differences were not statistically significant, indicating either larger cohorts are needed, or that the vaccine is not efficacious. The authors should explain, given the data, the advantage of the vaccine candidate.
Ramamoorthy et al show significance with their vaccine targeting multiple N. caninum antigens. Do the authors intend on including other antigens to get better immunity?
The manuscript could be further strengthened by included characterization of the immune response in terms of CD4, CD8, and antibody generation. Also, comparison of non-pregnant to pregnant subjects, in terms of immune response, could provide insight into protective mechanism.
Specific comments:
Line 99, suggest rewording. "Procedures that may result in pain or distress were performed…" rather than "Painful animal experiments in this study were…"
Line 135 Route of exposure? IN, IM, IP, oral gavage?
How does vaccination help establish the model?
Line 142, replace "freshly died" with "mice that succumbed to infection…."
Line 148 "lysis" instead of lyses' "parasites'" instead of parasites
Line 163 "mice" instead of "mouse"
Line 209 'offspring' instead of "offsprings"
It's not clear which groups are not significant.
Isn't the goal, sterilizing immunity or is reducing disease burden sufficient to reduce economic impact?
What is meant by "eating behavior of by dam"? Cannibalization or other?
ND detectable number was calculated or not observed?
Author Response
Reviewer 2
- The authors present a vaccine candidate to prevent vertical transmission of N. caninum. N. caninum infection of cattle results in nearly $1.3b annual loss to the cattle industry and is thus an economically important pathogen. Vaccines are necessary to prevent disease. The authors show in a murine model, reduction in pathogen burden weight loss, and pup survival, however the differences were not statistically significant, indicating either larger cohorts are needed, or that the vaccine is not efficacious. The authors should explain, given the data, the advantage of the vaccine candidate.
Author’s response:
In this study, we have evaluated the protective potential of NcGRA6 in pregnant mouse model by assaying survival rates in dams and pups, the parasite burden in different organs in pups and dams, body weights and clinical scores of dams and birth rates. These parameters are frequently tested in similar studies. According to our obtained data, survival rate of pups was statistically significant in NcGRA6-immunized mice against PBS or GST control groups in trial 1 and trial 2. Other data also showed significant difference in clinical score and body weight changes of dams where they were lower in NcGRA6-immunized groups than control ones. Result of parasite burden tended to be lower in NcGRA6-Immunized group than controls while there was not statistically significant difference.
These data are elaborately explained in results section (lines 190-272). Additionally, they have been briefly explained and discussed in discussion section (lines 303-310).
In addition, we added new information regarding the data explanation and advantages of mouse model, vaccine candidate and NcGRA6 as follows;
“Mouse models are highly beneficial in recognition of the key effector molecules of innate and adaptive immune responses against N. caninum infection. Indeed, using of different mouse models has greatly enhanced our understanding of N. caninum-host interactions and assisted in discoveries of potent vaccine candidates [5,6,16]” (lines 285-288).
“Herein, we demonstrated the high capacity of NcGRA6 in the reduction of vertical transmission of N. caninum in mice evidenced in higher survival rates and lower parasite burden in surviving offspring and dams compared with control groups. Some newborn mice were died from NcGRA6+GST-immunized group and PCR measurements revealed their positivity to N. caninum infection. However, the significant higher survival rate and lower parasite burden in the brain of surviving offspring indicate the adequacy of rNcGRA6 immunization in minimizing the vertical transmission.
In spite of using different mouse strains and N. caninum isolates, most of studies revealed that no vaccine candidates could induce sterile immunity or completely prevent the vertical transmission from dams to offspring especially if subunit vaccines were used.” (lines 303-313).
“However, because of safety, ease of preparation and affordability, subunit vaccines are advantageous than other kinds of vaccine. Although NcGRA6-conferred protection is partial in this study, this protective effect can be improved by several strategies such as using appropriate adjuvants or combination with other potent antigens.” (lines 326-329).
- Ramamoorthy et al show significance with their vaccine targeting multiple N. caninum antigens. Do the authors intend on including other antigens to get better immunity?
Author’s response:
Yes, we have a future plan to test recombinant protein of NcGRA6 as vaccine candidates after combination with other potent vaccine antigens or formulation with appropriate adjuvants as indicated in our revised manuscript as follows;
In introduction section;
“Indeed, vaccination of pregnant mice with NcGRA6 alone demonstrated promising results for protection against vertical transmission of neosporosis.” (lines 98-99)
In discussion section;
“Although NcGRA6-conferred protection is partial in this study, this protective effect can be improved by several strategies such as using appropriate adjuvants or combination with other potent antigens.” (lines 327-329)
- The manuscript could be further strengthened by included characterization of the immune response in terms of CD4, CD8, and antibody generation.
Author’s response:
We thank reviewer 2 for this important point. Actually, this study is a complementary study to our previously published study by Fereig et a1. 2019 (reference no. 21) which investigated the role recombinant protein of NcGRA6 as a vaccine candidate in non-pregnant mouse model. In such study, the immunization with NcGRA6 induced highly significant protection when compared to control groups (NcGRA6; 91.7%, GST; 25%, PBS; 16.7%). The mechanism of protection was also investigated comprehensively using multiple and various assays and markers. We found that solely NcGRA6 can trigger robust immunomodulation as evidenced in IFN-γ and IgG1 productions. In the current study, we have utilized the same antigens, immunization regimen (three immunization times within 2 weeks interval), same mouse sex and strain (female BALB/c), dose and route of administered antigen (25 pmol, subcutaneous) for recalling stimulation assay. Thus, our goal was focused primarily on assessment of the protective efficacy of recombinant NcGRA6 in pregnant mouse model particularly when similar tendencies of antibody responses after the immunization were also obtained between our current and published paper (reference No. 21). Because these data were not included in the previous version of our manuscript, we added results of antibody response in supplemental file.
We added this information in the revised version of our manuscript as follows;
“………Both studies have utilized the same immunization regimen (three immunization times within 2 weeks interval), same mouse sex and strain (female BALB/c), dose and route of administered antigen (25 pmol, subcutaneous) for recalling stimulation assay [21]. In addition, similar levels and tendencies for specific anti-NcGRA6 IgG1 and IgG2a were observed in our present study (see Figure S1 in Supplemental information file) and previous one [21]. Thus, we expect a potent role of IFN-γ and specific antibody to NcGRA6 in protection induced by NcGRA6 immunization in our current pregnant model.” (lines 357-363)
- Also, comparison of non-pregnant to pregnant subjects, in terms of immune response, could provide insight into protective mechanism.
Author’s response:
We have already added some information regarding pregnant and non-pregnant models either for cattle or mice as shown in discussion section (lines 330-341).
In addition, we have expanded this part in discussion section as follows;
“Conversely, several other studies reported the favorable effect of IFN-γ in normal pregnancy development. In case of Neospora challenge infection, many vaccine antigens induce IFN-γ production in re-stimulation assay of spleen cells while these antigens conferred high protection in the vertical transmission model [12,20,32]. In a similar effect but in the non-infected mice model, some reports revealed the necessity of IFN-γ for normal gestational development in mice [33,34]. These findings reflect the intricate immunomodulation during pregnancy, and the distinct effect of IFN-γ depending on the stage of pregnancy and site of operation. To maintain the pregnancy and control the parasite in dams, cell-mediated immunity at the site of infection is critical. Killing the intracellular N. caninum at the early stage of the infection may regulate the inflammatory reaction at a minimum level.” (lines 342-351).
Specific comments:
- Line 99, suggest rewording. "Procedures that may result in pain or distress were performed…" rather than "Painful animal experiments in this study were…"
Author’s response:
We have revised as requested as follows;
“Procedures that may result in pain or distress of mice in this study were performed under general anesthesia with isoflurane.” (lines 104-105).
- Line 135 Route of exposure? IN, IM, IP, oral gavage?
Author’s response:
We indicated the methods of immunization and challenge infection in method section as follows;
“All female mice were immunized via subcutaneous inoculation with recombinant proteins of NcGRA6+GST (25 pmol), GST (25 pmol) or PBS alone (100 µL) at three times by 14-day intervals.” (lines 140-142).
“At 4 weeks post-vaccination, only pregnant dams were challenged intraperitoneally on the same day with 1 × 105 tachyzoites of N. caninum (Nc-1) at day 10 of gestation.” (lines 145-147).
- How does vaccination help establish the model?
Author’s response:
“vertical transmission model, immunization, mating and challenge infection regimen was performed and optimized as previously described by several publications [ref. No. 12,13,14 in current manuscript].”
- Line 142, replace "freshly died" with "mice that succumbed to infection…."
Author’s response:
Done (line 150)
- Line 148 "lysis" instead of lyses' "parasites'" instead of parasite
Author’s response:
Revised (line 156 for lysis, and line 159 for parasites)
- Line 163 "mice" instead of "mouse"
Author’s response:
Replaced (line 170)
- Line 209 'offspring' instead of "offsprings"
Author’s response:
Done (throughout the text)
- It's not clear which groups are not significant.
Author’s response:
We specified the groups (PBS and GST) (line 223)
- Isn't the goal, sterilizing immunity or is reducing disease burden sufficient to reduce economic impact?
Author’s response:
The goal of this study is to control of vertical transmission of N. caninum from dams to pups and reduce severity in dams. Because it is difficult to completely inhibit the transmission, significant reduction of transmission is usually considered as a promising outcome for vaccination in pregnant mouse model. We have illustrated this information in our manuscript as follows;
“However, because of safety, ease of preparation and affordability, subunit vaccines are advantageous than other kinds of vaccine. Although NcGRA6-conferred protection is partial in this study, this protective effect can be improved by several strategies such as using appropriate adjuvants or combination with other potent antigens.” (lines 326-329)
- What is meant by "eating behavior of by dam"? Cannibalization or other?
Author’s response:
We have changed from “eating behavior by dam” to “cannibalistic behavior by dam” (line 230)
- ND detectable number was calculated or not observed?
Author’s response:
We meant by ND No detectable number was observed.
It is revised in the text (line 235)

Round 2
Reviewer 2 Report
The authors have presented a revised manuscript describing vaccine efficacy of nCGRA6 in the mouse model of Neospora caninum. The authors have addressed all scientific concerns.